# Tocopherol and Tocotrienol Content in the Leaves of the Genus *Hypericum*: Impact of Species and Drying Technique

**DOI:** 10.3390/plants14071079

**Published:** 2025-04-01

**Authors:** Ieva Miķelsone, Elise Sipeniece, Dalija Segliņa, Paweł Górnaś

**Affiliations:** Institute of Horticulture, Graudu 1, LV-3701 Dobele, Latvia; ieva.mikelsone@lathort.lv (I.M.); elise.sipeniece@lathort.lv (E.S.); dalija.seglina@lathort.lv (D.S.)

**Keywords:** St. John’s wort (*Hypericum perforatum* L.), photosynthetic, herb, ornamental plant, lipophilic bioactive compound, vitamin E, tocol

## Abstract

α-Tocopherol (α-T) predominates in photosynthetic tissues, while tocotrienols (T3s) are reported very rarely. The genus *Hypericum* stands out as one of the few exceptions. Given the potential health benefits associated with tocotrienols, sourcing them from natural origins is of interest. The proper selection of plant material and the drying conditions are crucial steps in this process. Therefore, in the present study, we investigated the effects of four different drying techniques (freeze-drying, microwave–vacuum-, infrared oven and air-drying) on the tocochromanol content in leaves of three *Hypericum* species: *H. androsaemum*, *H. pseudohenryi*, and *H. hookerianum* and one hybrid *H. × inodorum*. The total tocochromanol content in the freeze-dried leaves harvested in September was 68.1–150.6 mg/100 g dry weight. α-T constituted 66.7–85.9% (*w*/*w*), while tocotrienols constituted 13–32% (*w*/*w*). *H. pseudohenryi* was characterized by the lowest tocotrienol content, while *H. androsaemum* and *H. hookerianum* had the highest, with δ-T3 and γ-T3, respectively, being predominant. Tocotrienols were more stable during drying than α-T. The greatest decrease in α-T content was observed during air-drying in the presence of sunlight, with a 27% difference compared to the absence of sunlight. The species and harvest time are factors that more strongly affect the tocotrienol content in the *Hypericum* leaves than the selected drying method.

## 1. Introduction

The genus *Hypericum* belongs to the Hypericaceae family, which is interchangeably used with the Clusiaceae family (Hypericaceae = Clusiaceae) according to the Angiosperm Phylogeny Group (APG) system (1998 version) [1]. Since 2003, the APG system has recognized Hypericaceae Juss. (1789) and Clusiaceae Lindl. (1836) as two separate families [2]. Despite being classified into two distinct families, studies reveal a phylogenetic connection between the genera *Hypericum* (Hypericaceae) and *Clusia* (Clusiaceae): notably, in their relatively high leaf tocotrienol content—a phenomenon rarely observed in green plant tissues. This similarity highlights a shared characteristic that transcends their current taxonomic separation, suggesting deeper biochemical affinities between these genera [3]. *Hypericum* spp. includes herbs scarcely exceeding 20 cm high and small trees up to 10 m tall. *Hypericum* has an almost worldwide distribution, apart from extreme cold regions and dry regions. Over 25% of the species of genus *Hypericum* are popular in cultivation, mainly for ornamental and aesthetic reasons, but also for medical purposes (as a source of curative substances) [4]. According to estimates made in 2015/2016, the Hypericaceae family consists of six genera and 590 species [5]. The number is certainly higher now, especially considering the discovery of the species *Hypericum liboense* M.T.An and T.R.Wu in Guizhou, China [6].

The Greeks used *Hypericum* in a spiritual way. This practice has become associated in the Christian calendar with the feast of St. John the Baptist, hence its vernacular name of St. John’s wort. *Hypericum* was traditionally used as an external ointment for treating wounds in orthodox (Western) medicine [4]. Over the past few decades, interest has grown in using *Hypericum* extracts in traditional medicine to treat mild depression and to support anti-hyperglycemic, anti-inflammatory, anti-microbial, anti-oxidant, cytotoxic, hepatoprotective, and other activities [7]. In the West, the use of *Hypericum* spp. as a source of biologically active substances is nearly confined to *H. perforatum*, while, in other parts of the world, other species are used as well [4]. Although species of the genus *Hypericum* have been thoroughly studied and the plants have been recognized as rich in biologically active compounds such as naphtodianthrones, acylphloroglucinols, and polyphenols [8,9], currently, only the *Hypericum* genus is distinguished by its relatively high concentrations of tocotrienols in its leaves [3,10] and the inflorescences of wild *H. perforatum* [11,12,13]. *H. perforatum* is the most widely recognized medicinal plant [9], and it is traditionally used for diverse biological activities, especially for treating mild to moderate depression. Its primary bioactive compound, hyperforin, is believed to play a significant role in its antidepressant effects. While the exact mechanism of action is not fully understood, recent research suggests that it may involve the modulation of the gut microbiota [14]. The composition of the most widely investigated species of genus *Hypericum* (*H. perforatum*) may still conceal previously undiscovered rare bioactive compounds that are of great importance in understanding the medicinal properties of this plant, e.g., tocotrienols, mainly δ-T3 [11,12,13,15]. While hydrophilic compounds in general seem well studied [9], we know relatively little about lipophilic phytochemicals in the genus *Hypericum*, even when they are the subject of research. For instance, in four Tunisian *Hypericum* species—*H. perforatum*, *H. tomentosum*, *H. perfoliatum* and *H. ericoides*—the composition of fatty acids and tocopherols was investigated. However, the presence of tocotrienols, along with tocopherols, was not considered, despite the earlier discovery of δ-T3 in *H. perforatum* plants [16]. This shows that, despite several decades of research, there are still numerous undiscovered bioactive substances in the genus *Hypericum*, especially in ornamental species, which may have important applications in pharmacy/medicine, e.g., tocotrienols. Tocotrienols may have superior or different bioactive functions in comparison to tocopherols. Tocotrienols may have rejuvenating effects on pre-senescent and senescent primary cells [5]. Therefore, obtaining tocotrienols from sustainable/natural sources is important.

Natural drying methods, such as shade and wind drying, are the most commonly used techniques due to their simplicity and low costs. Additionally, they are environmentally sustainable. However, these methods offer limited control over the quality of the dried product and the drying rate. Conventional oven drying provides greater control over the process of drying the leaves [17], but it has its own set of drawbacks, including high energy expenditure and the potential degradation of product quality due to excessive heat. At the other end of the spectrum is freeze-drying, a method known for its ability to preserve sensitive nutrients. An intermediate solution appears to be microwave–vacuum drying [18], which offers a balance between efficiency and the preservation of thermolabile compounds. While many studies focus on minimizing the loss of hydrophilic bioactive compounds, such as polyphenols [19,20], there is limited research on lipophilic compounds, such as tocochromanols in leaves [18]. Meanwhile, the drying method is another critical factor affecting the final content of lipophilic bioactive compounds such as tocopherol, plastochromanol-8 and carotenoids [18]. In this study, these methods were tested to better understand how environmental factors, particularly ozone exposure during sun and air-drying, affect plant-derived compounds such as tocochromanols. Due to the limited number of natural sources of tocotrienols, there is a lack of studies on the stability of tocotrienols during drying. Hypothetically, due to unsaturated isoprenoid side chains with three carbon–carbon double bonds in tocotrienols, as compared to the saturated side chains for tocopherols, we can expect a higher rate of susceptibility to the loss of tocotrienols in comparison to tocopherols during drying. There are no studies on the effect of drying leaves on tocotrienols content.

Therefore, understanding how different drying methods affect the content of tocotrienols in the leaves of *Hypericum* species is crucial for optimizing the processing of this medicinal plant to minimize the losses of these lipophilic bioactive phytochemicals. The present study aims to investigate the effects of four different drying techniques on the tocochromanol content in the leaves of three *Hypericum* species, *H. androsaemum*, *H. pseudohenryi* and *H. hookerianum*, and one fertile hybrid of *H. androsaemum* and *H. hircinum* (*H. × inodorum*), with the goal of maximizing tocotrienol retention.

## 2. Results and Discussion

### 2.1. Tocochromanol Profile in the Freeze-Dried Leaves of the Genus Hypericum

The saponification protocol, used to prepare samples for tocochromanols determination, is the most commonly used method as it achieves the highest recovery of these lipophilic antioxidants compared to other methods. This protocol enhances extractability from the plant matrix and releases bound tocochromanols, such as esters, into free compounds [21,22]. However, some losses, particularly of tocotrienols, have been observed when using this protocol [22]. To reduce the quantity of used harmful mixtures of *n*-hexane and ethyl acetate from 50 mL [23] to 7.5 mL, we adopted the semi-micro saponification protocol proposed by Slavin and Yu [24], which was later validated on apple seeds [25]. The toxicity of *n*-hexane can be reduced by the substitution of the less toxic *n*-heptane, albeit at a higher solvent cost [26]. One of the more innovative and environmentally friendly approaches appears to be a protocol using only ethanol as a solvent, combined with ultrasound-assisted extraction. This method has been shown to be somewhat comparable to the saponification protocol, as demonstrated in studies on cranberry seeds [27], grape seeds [22], and *H. perforatum* inflorescences [12], particularly regarding tocotrienol recovery. In contrast, greater differences were noted for tocopherols, especially α-T [12,22].

In the leaves of all examined *Hypericum* samples, three primary tocochromanols were identified: α-T, γ-T3, and δ-T3. The content of γ-T was low, while the amounts of β-T, δ-T, α-T3, and β-T3 were either trace or undetectable. Figure 1 shows an example of a chromatogram obtained from the leaves of *H. androsaemum*, along with four tocopherol and four tocotrienol standards.

The content and proportion of tocopherols and tocotrienols in leaves of three *Hypericum* species and one hybrid (*H. androsaemum*, *H. pseudohenryi*, *H. hookerianum* and *H. × inodorum*, respectively) are shown in Figure 2 and Figure 3.

The leaves of *H. pseudohenryi* were characterized by the lowest tocopherol and tocotrienol content (59.25 and 8.83 mg/100 g dw, respectively), while *H. androsaemum* had the highest (117.30 and 33.31 mg/100 g dw, respectively). Leaves of *H. × inodorum* were characterized by slightly lower content of tocopherols and tocotrienols (about 5% and 20%, respectively) compared to *H. androsaemum*. In turn, the leaves of *H. hookerianum* contain tocopherol levels similar to those of *H. pseudohenryi* and tocotrienol levels comparable to those of *H. × inodorum* and *H. androsaemum* (about a 5% difference) (Figure 2). The presence of tocochromanols in medicinal plants was reported by Inoue et al. in 2012 [16], including *H. perforatum*, where the occurrence of α-T and δ-T3 was identified. The qualitative determination was made using mass spectrometry (MS); however, no quantitative determination was performed [16]. In contrast, the presence of tocotrienols was not taken into account (tocotrienol standards were not applied) in the study of four Tunisian *Hypericum* species: *H. ericoides*, *H. perfoliatum*, *H. perforatum*, and *H. tomentosum* [28]. In turn, relatively high concentrations of tocotrienols have been reported in the leaves of several *Hypericum* species: *H. aegypticum* (12.54 mg/100 g dw), *H. calycinum* (8.51 mg/100 g dw), *H. empetrifolium* (12.45 mg/100 g dw), *H. lancasteri* (24.27 mg/100 g dw), *H. olympicum* (52.22 mg/100 g dw), *H. perforatum* (13.29 mg/100 g dw), and *H. xylosteifolium* (0.61 mg/100 g dw) [3], and *H. calycinum* (5.84 mg/100 g dw), *H. densifolium* (16.88 mg/100 g dw), *H. frondosum* (12.08 mg/100 g dw), *H. hircinum* (0.97 mg/100 g dw), *H. hookerianum* (15.93 mg/100 g dw), *H. kalmianum* (5.52 mg/100 g dw), *H. olympicum* (15.08 mg/100 g dw), *H. prolificum* (22.9 mg/100 g dw), and *H. xylosteifolium* (0.78 mg/100 g dw), as well as two hybrids: *H. × moserianum* (1.19 mg/100 g dw) and *H* × ‘Rowallane’ (20.49 mg/100 g dw) [10]. Tocopherols are the main tocochromanols in plant tissues, mainly α-T in leaves and γ-T in seeds, while the presence of tocotrienols is scarce [29]. Therefore, the abundance of α-T in the leaves of the genus *Hypericum* is unsurprising, as α-T is well-established as the primary tocochromanol in leaves regardless of the plant species, with some exceptions, e.g., *Kalanchoe daigremontiana* and lettuce, where γ-T was a reported as a primary tocochromanol [30]. This study confirms the thesis about α-T domination while, at the same time, shedding light on specific phylogenetic relationships within the *Hypericum* genus, which are associated with the presence of relatively high concentrations of tocotrienols in the leaves of its species, as reported elsewhere [3,10]. Observed similarities in the tocochromanol profile and content between *H. androsaemum* and *H. × inodorum* (section *Androsaemum*) as well as *H. pseudohenryi* and *H. hookerianum* (section *Ascyreia*) may be due to their genotypic similarities (belonging to a particular section). It has been reported that chemotaxonomy could be used as a useful tool for plant classification based on their secondary metabolites [31]. It has been demonstrated that the proper solution may involve applying a chemotaxonomic tool in the search for new plant resources that are rich in tocotrienols, more precisely, in seeds from the Apiaceae family [32] and plant oils from species in families such as Arecaceae and Poaceae (monocots), as well as dicots such as Ericaceae and Vitaceae [33]. Similar statements can be made based on the current study in the leaves of the *Hypericum* genus. However, both this and previous studies require additional research to solidify these phylogenetic relationships, given the limited scope of the species analyzed to date. More species need to be studied in the future to make this statement more reliable. Additionally, it is essential to consider other potential variables, in addition to the influence of the species, which may influence the content of tocochromanols in leaves, such as seasonal fluctuations (harvest time) [18,34], leaf size [34], plant gender [18], environmental conditions [35], and plant age [36]. Comprehending these factors is essential for evaluating potential future industrial applications and profitability.

The lowest percentage of tocotrienols was recorded in the leaves of *H. pseudohenryi* (13%), while the highest was found in *H. hookerianum* (32%). In both species, γ-T3 was the predominant tocotrienol, comprising 10% and 25% of the total tocotrienols, respectively. The leaves of *H. androsaemum* and *H. × inodorum* contained a similar proportion of tocotrienols (22% and 19%, respectively), with a slight 1–2% predominance of δ-T3 over γ-T3 (Figure 3). The homologue α was the dominant one among tocopherols, while δ and γ were predominant among tocotrienols in investigated *Hypericum* samples in the present study. The dominance of homologue δ in tocotrienols and α in tocopherols, as observed in the leaves of *H. androsaemum* and *H. × inodorum*, has also been reported in *H. perforatum* and some *Clusia* species [3]. In contrast, the dominance of γ-T3 and α-T, noted in the leaves of *H. pseudohenryi* and *H. hookerianum*, aligns with findings for most *Hypericum* species [3,10]. The prevalence of δ-T3 from all tocotrienols is worth highlighting due to its rarity; however, studies indicate that it is a characteristic tocotrienol in *H. perforatum* [12,16], with inflorescences that are especially rich in δ-T3 [12]. The tocotrienol content in the inflorescences of wild St. John’s wort is comparable to that found in palm (*Elaeis guineensis* Jacq.) oil and is approximately ten times lower than the tocotrienol content in annatto (*Bixa orellana* L.) seeds [15]. There are no reports regarding the tocochromanol profile in *H. androsaemum*, *H. × inodorum*, and *H. pseudohenryi*. In turn, in the leaves of *H. hookerianum*, the same tocochromanol profile was reported [10], although the concentrations were approximately half of those observed in the current study. This could be a result of several factors, e.g., plant size [34]. Tocotrienols are typically scarce in the green tissues of higher plants; trace amounts of α-T3 have been detected in spruce needles (genus *Picea*) [37] and 4 to 12 mg/100 g dw of two tocotrienols (β-T3 and γ-T3) in the leaves of the *Vellozia gigantea* [34]. The concentration of tocotrienols in leaves can be increased through transgenic manipulation [38]. Both tocopherols and tocotrienols are synthesized through similar biochemical pathways: primarily, the shikimate pathway and the 2-C-methyl-D-erythritol 4-phosphate (MEP) pathway. The initial step in tocochromanol biosynthesis involves the conversion of L-tyrosine into homogentisic acid (HGA) through enzymatic reactions, including transamination by tyrosine aminotransferases (TATs). HGA serves as a precursor for both tocopherols and tocotrienols. The key difference in their biosynthesis lies in the side chains attached to the chromanol ring. Tocopherols have a saturated phytyl side chain derived from phytyl pyrophosphate (PPP), while tocotrienols possess an unsaturated side chain with three double bonds, originating from geranylgeranyl pyrophosphate (GGPP). This structural distinction is crucial as it influences their biological activity and antioxidant properties [39]. Limited reports exist on tocotrienols in photosynthesizing organs, and research on their functions within these tissues is still scarce. Despite both tocopherols and tocotrienols exhibiting antioxidant properties, their roles may differ significantly in green tissues. Future studies should focus on elucidating these differences and exploring how environmental factors influence tocochromanol profiles across various *Hypericum* species.

### 2.2. Impact of the Drying Method, Species, and Their Interaction on Tocochromanol Content in the Leaves of the Genus Hypericum

The statistical analysis results of how the species and drying methods affect tocochromanol levels in *Hypericum* leaves were visualized using boxplots. These plots highlighted the distribution of three primary tocochromanols (α-T, γ-T3, and δ-T3) (Figure 4) and two secondary ones (β-T and γ-T) (Figure 5). The species factor exhibited a statistically significant influence (*p* < 0.0006) on α-T levels among the *Hypericum* samples tested, with the exception of the pairs *H. androsaemum* and *H. inodorum*, as well as *H. pseudohenryi* and *H. hookerianum*, which showed no significant differences. A similar pattern was observed for β-T (*p* < 0.00001). However, for γ-T3, δ-T3, and γ-T, all species demonstrated statistically significant differences from one another (*p* < 0.03). While the species had a significant impact on the content of each tocochromanol, the drying method had a notable impact only on α-T and β-T (*p* < 0.05). In both cases, air-drying resulted in the lowest content. It is worth noting that no statistical differences were found in α-T content among the other drying methods, nor for β-T when comparing lyophilization with the other methods. The microwave–vacuum, freeze-drying, and infrared methods were found to be the most effective in preserving α-T. Air-drying reduced α-T content by two to three times compared to the other drying methods. Conversely, the drying method had no effect on tocotrienol content, which is noteworthy. This finding is particularly valuable for future production efforts related to the extraction and isolation of tocotrienols from *Hypericum* leaves. Due to the low content of β-T and γ-T (1–2% total tocopherols), the influence of biological replicates and the measurement error, it is scientifically inappropriate to discuss the influence of the drying method on the content of those tocopherols. Similarly, an advantage of the microwave–vacuum and freeze-drying methods over conventional drying on tocochromanol preservation in sea buckthorn (*Hippophae rhamnoides* L.) leaves was reported [18]. Another study reported a decrease of around 25% in α-T content in kale (*Brassica oleracea* L. var. *acephala*) leaves dried using hot (55 °C) air for 5 h in comparison to freeze-drying [40]. Given the rarity of reports on the presence of tocotrienol in the leaves of other species, comparative data regarding the stability of tocotrienols during leaf drying are very scarce. This study marks the first instance where it has been unexpectedly found that tocotrienols exhibit greater stability than tocopherols, with α-T showing notably low stability, in leaves of the genus *Hypericum*.

### 2.3. The Impact of Sunlight and a One-Month Delay in Leaf Harvesting of H. androsaemum and H. × inodorum

The resulting significant differences in α-T content (and the lack thereof for tocotrienols) between the other drying methods and air-drying led us to conduct another air-drying experiment in the presence and absence of sunlight a month later. For this experiment, only the *H. androsaemum* and *H. × inodorum* leaves were tested. The additional study produced two important findings. First, the leaves of *H. androsaemum* and *H. × inodorum* harvested in October had a higher content of tocopherols and tocotrienols (approximately 10–20%) relative to leaves harvested in September. Second, drying with or without sunlight had a statistically significant effect (*p* = 0.0039) on α-T levels (Figure 6).

This experiment highlighted that sunlight has the biggest destructive impact on the α-T in the leaves during air-drying, leading to a reduction of about 27% under solar light compared to conditions lacking solar light over seven days. The species itself had no statistically significant impact on α-T content in *H. androsaemum* and *H. × inodorum*, consistent with the findings from the September collections. The content of the other two tocopherols, β-T and γ-T, in the leaves of *H. androsaemum* and *H. × inodorum* collected in October did not differ significantly between species or air-drying conditions. Air-drying with or without the presence of sunlight did not affect the γ-T3 and δ-T3 concentrations in the leaves of these two species, similar to the case of β-T and γ-T. However, these species exhibited statistically significant differences (*p* = 0.0104 and *p* = 0.0374) regarding the content of tocotrienol, similarly to observations made for the leaves collected in September. Notably, the lack of a significant sunlight effect on tocotrienol content is beneficial for maintaining their stability during the leaf-drying process. Furthermore, this study highlights the unique roles of α-T in comparison to other tocochromanols in *Hypericum* leaves, although these differences could not be fully explained by the present study. The distribution of tocotrienols across plant organs is less common than that of tocopherols, which are more prevalent in photosynthetic tissues. This suggests that the biosynthesis and accumulation of tocotrienols are more specialized than those of tocopherols [29]. Due to the limited number of reports on tocotrienols in photosynthesizing organs, research on their functions in these tissues is also scarce. The function of α-T is much better defined in green plant tissues. α-T is a lipophilic antioxidant that contributes to the protection of photosystem II against photodamage under environmental stress. It has been demonstrated that an RNAi line (vte5) with decreased expression of VTE5 and reduced levels of α-T protects against combined high-light and high-temperature stress via α-T production in tomato leaves [41]. The content of α-T exhibit significant variability, reflecting the plant’s adaptive response to environmental challenges and highlighting its capacity to withstand stress conditions [42].

The phenomenon of tocochromanol accumulation in leaves during plant development has been observed in both monocotyledonous and dicotyledonous plants. For instance, in the *H. rhamnoides* leaves collected in autumn (October), an increase in tocochromanol content of more than two times was observed compared to those collected in summer (June) [18]. The seasonal effect on the levels of both tocopherols and tocotrienols in the leaves has also been reported in *V. gigantea* [34]. The age of the leaf and its location on the plant play important roles in the content of tocochromanols [30,43].

## 3. Materials and Methods

### 3.1. Reagents

Ethanol, methanol, ethyl acetate, *n*-hexane (HPLC grade), pyrogallol, sodium chloride, and potassium hydroxide (reagent grade) were purchased from Sigma-Aldrich (Steinheim, Germany). Ethanol (96.2%, *v*/*v*) for the saponification of the leaf samples was received from SIA Kalsnavas Elevators (Jaunkalsnava, Latvia). Standards of tocopherol homologues (α, β, γ, and δ) (>98%, HPLC) were obtained from Extrasynthese (Genay, France), while tocotrienol homologues (α, β, γ, and δ) (>98%, HPLC) were obtained from Cayman Chemical (Ann Arbor, MI, USA).

### 3.2. Plant Material

The seeds of *H. androsaemum* and *H. × inodorum* were provided by Plant World Seeds (Devon, UK), *H. pseudohenryi* by the Botanical Garden of Marie Curie-Sklodowska University (Lublin, Poland), and *H. hookerianum* by the Botanical Garden of the Faculty of Science, Masaryk University (Brno, Czech Republic). The seeds were sown in February/March 2021 in a greenhouse using peat KKS-U from LaFlora (Līvbērzes, Latvia). In April/May, the plants were replanted in bigger pots and, finally, in June/July, they were planted in the garden of the Institute of Horticulture, Dobele, Latvia (GPS location: N: 56°36′39″ E: 23°17′50″). The *Hypericum* plants were grown on clay soil (pH—7.6; organic matter—2.1%). A detailed characterization of the soil is presented in Appendix A. *Hypericum* plants were collected in the middle of September (three species and one hybrid) and October (one species, *H. androsaemum*, and one hybrid, *H. × inodorum*) 2024, in the morning hours, by cutting branches of 10–15 cm from the ground into plastic black bags (protecting them from sunlight and moisture losses). Each species was collected in three biological replicates by cutting random plants. Photographs of the plants under investigation are presented in Appendix A. The healthy/non-damaged leaves were manually removed from the branches and gently mixed in the large plastic bag (three plastic bags, one for each biological replicate). The meteorological conditions, comprising air temperature (°C) (Appendix A) and sunshine hours (hh) (Appendix A) for the experimental period of September–October 2024, are detailed in the Appendix A. The leaves from each bag were then divided into four equal portions (each weighing 50–100 g) for four different drying techniques. Each sample was fully processed on the same day, with the exception of freeze-drying (which was performed on subsequent days, each time with one biological replication), including harvesting, separation, and drying. Microwave drying was conducted three times (each time with one biological replication) on the same day due to its short drying duration, while the infrared and air-drying methods were performed simultaneously with three separate biological replicates, due to the longer drying time required for these methods. The drying conditions and equipment utilized are depicted in Appendix A. 

### 3.3. Drying of Hypericum Leaves

To assess the effect of the drying methods on tocochromanol content, four distinct drying techniques were employed: freeze-drying, microwave–vacuum drying, infrared oven drying, and air-drying in the partial presence of sunlight and dark. Air-drying was evaluated as it is a centuries-old technique that continues to be employed in the drying of various materials, including medicinal plants. Microwave–vacuum drying was employed due to its rapid drying capabilities and ability to effectively preserving sensitive bioactive compounds, serving as good an alternative to freeze-drying. Infrared oven drying is one more available option, which we were able to test during this study. Such approach allowed for a comprehensive evaluation of how different drying methods influence the stability of tocopherols and tocotrienols in Hypericum leaves. These methods encompassed all the available drying options accessible at the Institute of Horticulture.

#### 3.3.1. Freeze Drying

*Hypericum* leaves were frozen in −80 ± 2 °C for 2 h and freeze-dried using a FreeZone 4.5 L freeze-dry system (Labconco, Kansas City, MO, USA) at a temperature of −51 ± 1 °C under a vacuum of 0.055–0.065 mbar for 72 h. The dry leaves reached a moisture level of 5 ± 1%.

#### 3.3.2. Microwave–Vacuum Drying

Microwave–vacuum drying was optimized according to the sample weight and desired moisture content based on previous studies concerning the drying of sea buckthorn (*H. rhamnoides* L.) leaves, which did not contain tocotrienols [18]. *Hypericum* leaves were placed in a container designed for a microwave–vacuum dryer (Model Musson-1, Ingredient, Saint Petersburg, Russia), which rotated at a speed of 6 rotations per minute. The drier was programmed as follows: power of 4 magnetrons, a pressure difference of 60/80 mmHg, a temperature of 60 °C, and a drying time of 2 min 30 s. The dried leaves achieved a moisture level of 5 ± 2%.

#### 3.3.3. Infrared Oven Drying

*Hypericum* leaves were placed on special trays (400 × 450 mm) with holes, for free air flow, designed for a forced-air circulation conventional infrared drying cabinet US-12 (Sushilnoe Delo Co Ltd., Saint Petersburg, Russia). *Hypericum* leaves were dried at a temperature of 40 ± 5 °C with electrical power of 9 kW for 42 h. The dry leaves were reached moisture level of 8 ± 2%.

#### 3.3.4. Air-Drying (Part Presence of Sunlight and Dark)

Air-drying was performed twice: the first time with partial exposure to sunlight (18 to 25 September 2024) and the second time with both partial exposures to sunlight and in darkness (16 to 23 October 2024). The second experiment was conducted as a result of observations made during the first experiment (a significant decrease in α-T content in air-dried leaves with the partial presence of sunlight) to test the hypothesis regarding the influence of sunlight and only with on two species *H. androsaemum* and *H. × inodorum*. Briefly, the first time, the *Hypericum* leaves were placed on parchment paper on the ground in a spacious attic room with polycarbonate windows located only on the west side, exposed to direct afternoon sunlight, and left for seven days. The leaves were exposed to direct sunlight for 4 ± 1 h per day. The air temperature during this period ranged from 12 to 25 °C during the daytime and 8 to 12 °C at night. The temperature and sunlight duration were determined using data from a meteorological station (Appendix A). The dried leaves achieved a moisture level of 12 ± 2%.

The second time, the leaves harvested on the morning of 16 October were dried in the same place as the first time, with partial exposure to sunlight and in darkness for 7 days, as before. The leaves kept in darkness were placed in plastic boxes with bottom and side holes to facilitate air circulation. These boxes were covered with panels and positioned in the darkest area of the room to minimize exposure to sunlight. Due to the late autumn sun, the leaf samples were exposed to direct sunlight for approximately 3 ± 1 h per day. During this experiment, the temperature was lower, ranging from 0 to 12 °C during the daytime and from 0 to 11 °C at night.

All drying equipment used is detailed in the Appendix A.

#### 3.3.5. Powdering

For each sample, 10 ± 2 g of dried plant material was finely powdered using an MM 400 mixer mill (Retsch, Haan, Germany). The leaves were placed in stainless steel grinding jars with screw tops and milled under the following conditions: a frequency of 30 Hz for 30 s. The resulting powder had a final fineness of 5 µm (as specified by the manufacture) and was used directly for the extraction of the tocopherol and tocotrienol homologues, as described in Section 3.4, as well as for measuring the dry mass (determined gravimetrically). Any remaining powdered samples were transferred to polypropylene bags and stored at −18 °C.

### 3.4. Semi-Micro-Saponification Protocol

To extract tocochromanols from dried *Hypericum* leaves, the saponification protocol was chosen due to its ability to achieve the highest recovery of tocopherols and tocotrienols from plant material [21,22]. This protocol was conducted using a previously validated procedure [25], with slight modifications: absolute ethanol was replaced by 96.2% (*v*/*v*) ethanol to reduce expenses. This replacement does not affect tocochromanol recovery [10]. The employed procedure of saponification does not allow for differentiation between free, bound, and non-extractable tocochromanols [21,22].

### 3.5. Tocopherol and Tocotrienol Determination via RP-HPLC-FLD

The tocochromanol analysis was performed using reverse-phase high-performance liquid chromatography with a fluorescent light detector (RP-HPLC-FLD) on a HPLC Shimadzu Nexera 40 Series system (Kyoto, Japan) consisting of a pump (LC-40D pump), a degasser (DGU-405), a system controller (CBM-40), an auto-injector (SIL-40C), a column oven (CTO-40C), and a fluorescence detector (RF-20Axs). The chromatographic separation of the tocopherol and tocotrienol homologues was carried out on the Epic PFP-LB (pentafluorophenyl phase) column (PerkinElmer, Waltham, MA, USA) with the following parameters: particle morphology—fully porous; particle size—3 µm, column length—150 mm; and column ID—4.6 mm; secured with a guard column of the length—4 mm and ID—3 mm (Phenomenex, Torrance, CA, USA). The chromatography analysis was performed under the following isocratic conditions: mobile phase—methanol with water (91:9; *v*/*v*); flow rate—1.0 mL/min; column oven temperature—45 ± 1 °C; room temperature—21 ± 1 °C. The total chromatography runtime was 13 min. The identification and quantification were performed using a fluorescence detector at an excitation wavelength of 295 nm and an emission wavelength of 330 nm. The quantification was achieved based on the calibration curves obtained from the tocopherol and tocotrienol standards. To precisely determine the concentration of the received tocopherol and tocotrienol standards in ethanol, the stock solutions were appropriately diluted in ethanol to obtain an absorbance within the range of 0.2 to 0.5, ensuring linearity according to the Beer–Lambert law. The obtained standard solutions were determined spectrophotometrically using extinction coefficients for individual tocochromanols and the Beer–Lambert equation [44]. Details of the method of HPLC validation were provided earlier [45]. The method was characterized by good repeatability and reproducibility in the range of 2.5–9.0%

### 3.6. Statistical Analysis

Four *Hypericum* species, each with three biological replicates, were subjected to four different drying methods in September; two species were subjected to air-drying with exposure to sunlight and in darkness in October (*H. androsaemum* and *H. × inodorum*). This resulted in a total of 48 samples (4 × 3 × 4 = 48) in September and 12 samples (2 × 3 × 1 × 2 = 12) in October. Figure 2 and Figure 3 were obtained based on average results using Excel (Version 2302) Microsoft 365 Apps for enterprise (Redmond, WA, USA) software.

The statistical analysis aimed to evaluate the effects of two factors—‘species’ and ‘drying method’—on five dependent variables. Three were major (‘α-T’, ‘γ-T3’, and ‘δ-T3’) and two were minor (‘β-T’, and ‘γ-T’). Given the non-normal distribution of the data, statistical methods appropriate for non-normal and heteroscedastic data were employed. Specifically, the Scheirer–Ray–Hare test, a non-parametric analog of two-way analysis of variance (ANOVA), was used to independently assess the influence of each factor on the dependent variables, without requiring assumptions of normality. This test was conducted separately for each dependent variable, and test statistics and *p*-values were calculated for both factors (‘species’ and ‘drying method’). For dependent variables exhibiting significant effects in the Scheirer–Ray–Hare test (*p* < 0.05), post hoc pairwise comparisons were performed using the non-parametric Mann–Whitney U test to compare two independent groups. The results were corrected for multiple comparisons using the Bonferroni method to limit the risk of Type I errors. The results are presented as boxplots (showing medians) to illustrate the distribution of dependent variables across groups defined by the ‘species’ and ‘drying method’ factors. These visualizations facilitated the identification of patterns and significant differences between groups. Statistical analyses were conducted using the Python programming language (3.12.7 packaged by Anaconda, Inc., Austin, TX, USA) with the following libraries: ‘scipy.stats’ for statistical tests (Scheirer–Ray–Hare and Mann–Whitney U); ‘matplotlib’ and ‘seaborn’ for data visualization; and ‘pandas’ for the data processing and analysis.

## 4. Conclusions

The utility of a chemotaxonomic tool in identifying new plant species valuable in terms of tocotrienols, as shown earlier in leaves [3,10], seeds [32], and plant oils [33], was confirmed in the present study in the leaves of the genus *Hypericum*. This study demonstrates the phylogenetic relationships among the *Hypericum* genus, and the presence of relatively high tocotrienol levels in the leaves of *H. androsaemum*, *H. hookerianum*, *H. pseudohenryi*, and the hybrid *H. × inodorum*. However, more species need to be studied in the future to make this statement more reliable.

The species factor seems to exert a more pronounced influence on the levels of tocochromanols, especially tocotrienols, than the drying method. *H. pseudohenryi* exhibited the lowest tocotrienol content, while *H. androsaemum* and *H. hookerianum* had the highest levels, dominated by δ-T3 and γ-T3, respectively. The microwave–vacuum, freeze-drying, and infrared methods were found to be somewhat comparable in preserving tocochromanols during the drying of *Hypericum* leaves. Surprisingly, tocotrienols demonstrated greater stability during drying than α-T. Additionally, it was found that exposure to sunlight is a crucial factor that substantially decreases the levels of α-T, while having no significant effect on tocotrienols, during the drying. This may be attributed to the physiological functions of α-T in leaves. This phenomenon requires more studies. Moreover, in addition to the species, drying method, and exposure to sunlight, the harvest time is a crucial factor determining the tocopherol and tocotrienol content in the *Hypericum* genus.

Given the widely known health benefits of tocotrienols and the limited natural sources available, our study highlights *Hypericum* as a promising alternative plant material for tocotrienol isolation in temperate regions. The current research underscores the need for further investigations to fully utilize the potential of *Hypericum* species. Future studies should aim to enhance our understanding of the presence, function, and concentration changes of tocotrienols in leaves across the plant’s growth period. Additionally, these studies should develop sustainable extraction and purification methods to isolate tocotrienols from other phytochemicals, with the long-term objective of utilizing them in pharmaceutical and medical applications.

## Figures and Tables

**Figure 1 plants-14-01079-f001:**
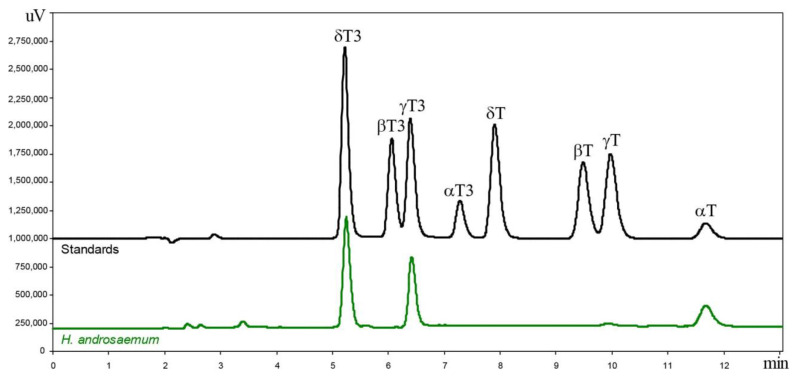
Chromatograms of the tocotrienol (T3) and tocopherol (T) homologues (α, β, γ, and δ): separation via RP-HPLC-FLD in the leaves of *H. androsaemum* (δ-T3, γ-T3, γ-T, and α-T) and standards (all eight tocochromanols).

**Figure 2 plants-14-01079-f002:**
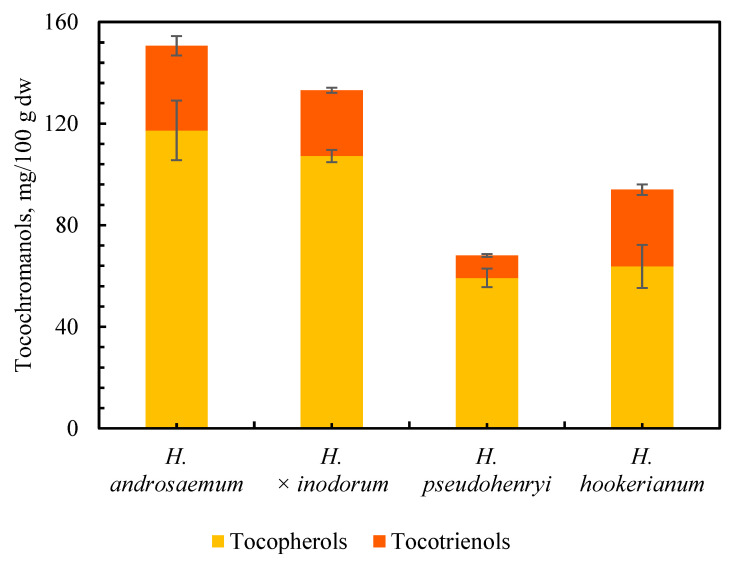
Contents (mg/100 g dw) of tocochromanols (tocopherols and tocotrienols) in the freeze- dried leaves of three *Hypericum* species and one hybrid, *H. androsaemum*, *H. pseudohenryi*, *H. hookerianum*, and *H. × inodorum*, respectively, collected in the middle of September 2024.

**Figure 3 plants-14-01079-f003:**
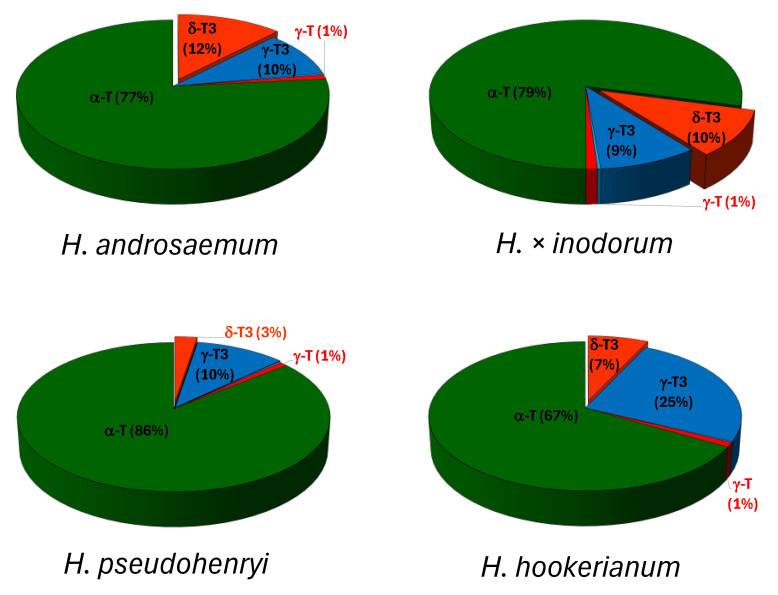
The average content proportion (%) of individual tocotrienol (T3) and tocopherol (T) homologues (α, β, γ, and δ) in the freeze-dried leaves of three *Hypericum* species and one hybrid, *H. androsaemum*, *H. pseudohenryi*, *H. hookerianum*, and *H. × inodorum*, respectively, collected in the middle of September 2024.

**Figure 4 plants-14-01079-f004:**
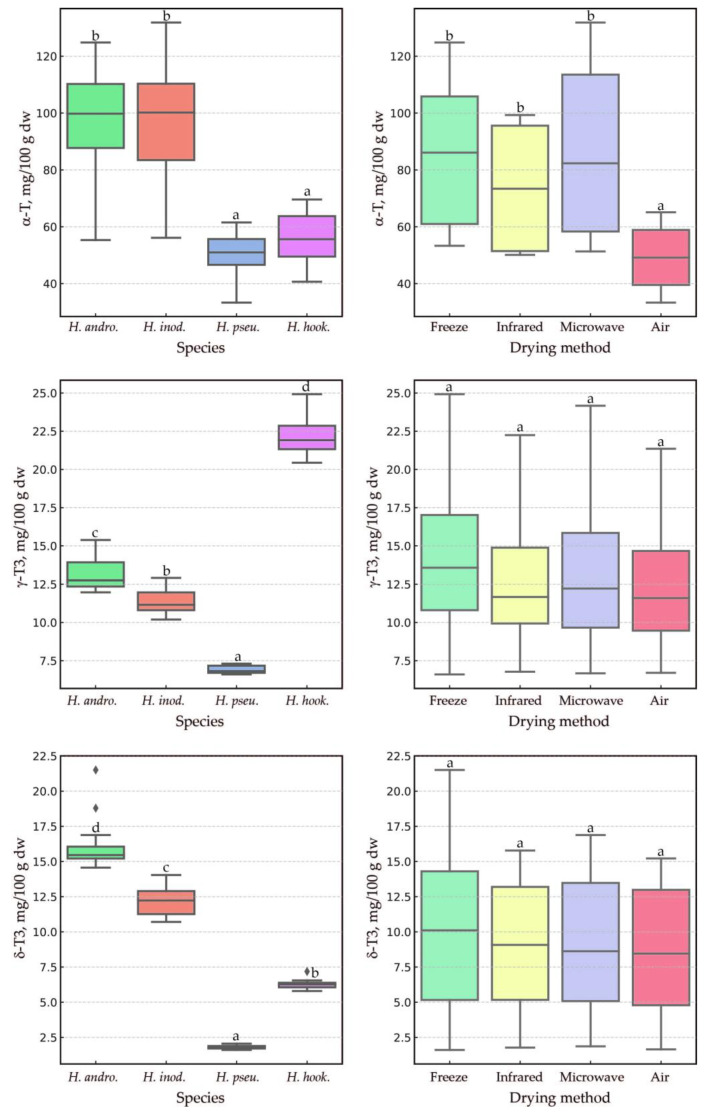
Boxplots illustrating the distribution of three major dependent variables (α-T, γ-T3, and δ-T3) in the leaves of three *Hypericum* species and one hybrid, *H. androsaemum*, *H. pseudohenryi*, *H. hookerianum*, and *H. × inodorum*, respectively, harvested in September, across groups defined by the ‘species’ and ‘drying method’ factors. Different letters indicate statistically significant differences at *p* < 0.05. T, tocopherol; T3, tocotrienol; dw, dry weight; *H. andro.*, *H. androsaemum*; *H. inod.*, *H. × inodorum*; *H. pseu.*, *H. pseudohenryi*; *H. hook.*, *H. hookerianum*.

**Figure 5 plants-14-01079-f005:**
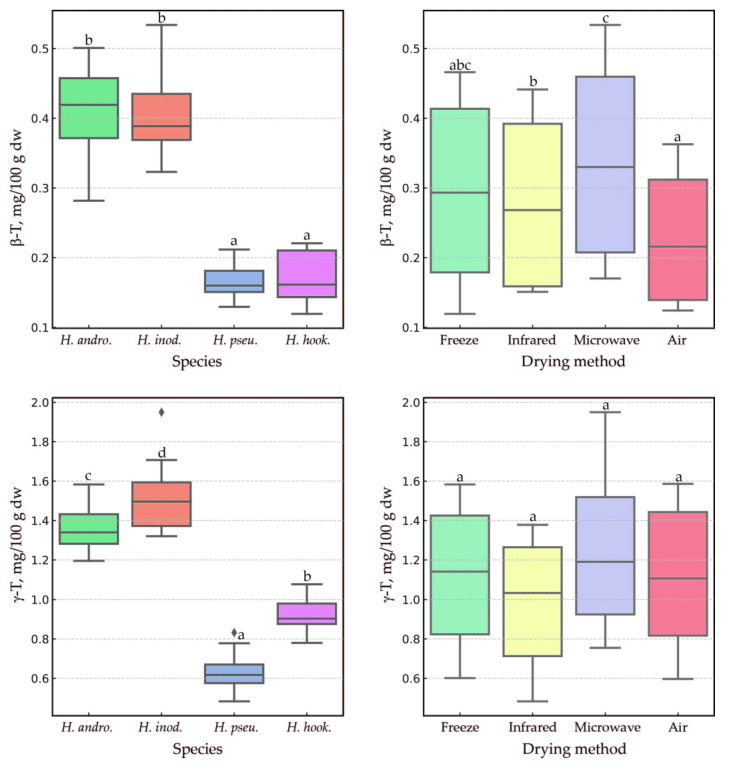
Boxplots illustrating the distribution of two minor dependent variables (β-T and γ-T) in the leaves of three *Hypericum* species and one hybrid, *H. androsaemum*, *H. pseudohenryi*, *H. hookerianum*, and *H. × inodorum*, respectively, harvested in September, across groups defined by the ‘species’ and ‘drying method’ factors. Different letters indicate statistically significant differences at *p* < 0.05. T, tocopherol; dw, dry weight; *H. andro.*, *H. androsaemum*; *H. inod.*, *H. × inodorum*; *H. pseu.*, *H. pseudohenryi*; *H. hook.*, *H. hookerianum*.

**Figure 6 plants-14-01079-f006:**
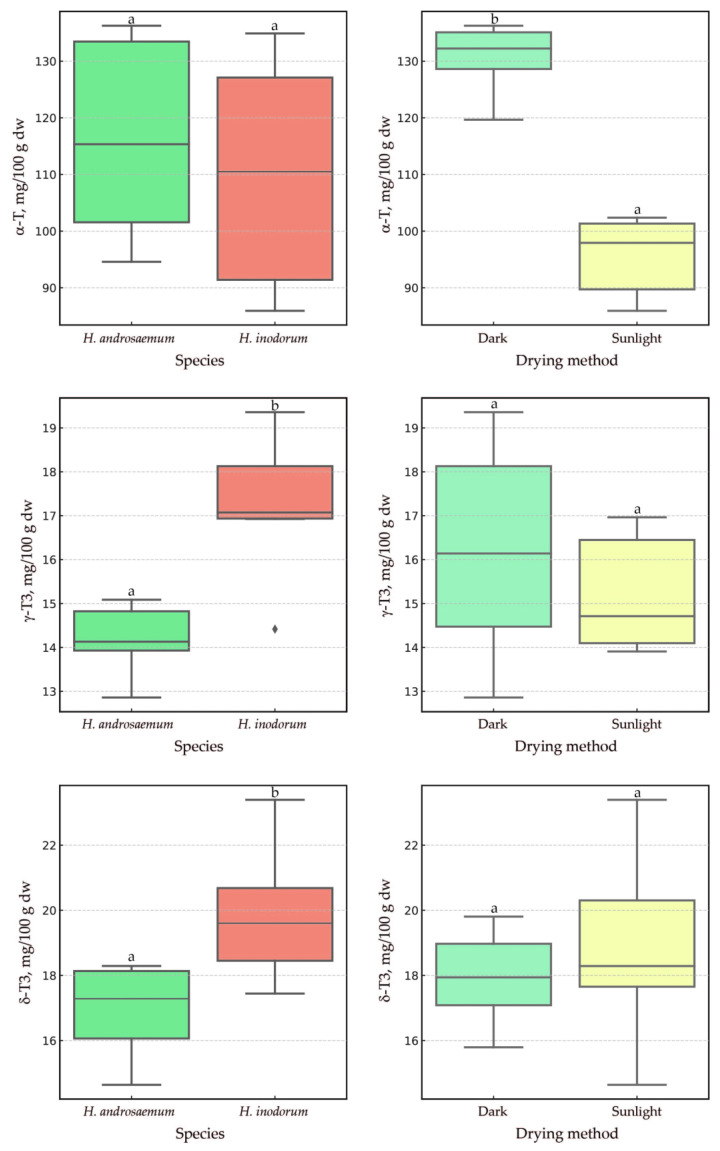
Boxplots illustrating the distribution of three major dependent variables (α-T, γ-T3, and δ-T3) in the leaves of *H. androsaemum* and *H. × inodorum*, harvested in October, across groups defined by the ‘species’ and ‘drying method’ factors. Different letters indicate statistically significant differences at *p* < 0.05. T, tocopherol; T3, tocotrienol; dw, dry weight.

## Data Availability

The data used to support the findings of this study are available in the Appendix A and from the corresponding author upon request.

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
