# Peer review of "Tocopherol and Tocotrienol Content in the Leaves of the Genus Hypericum: Impact of Species and Drying Technique"

_plants, 2025, doi:10.3390/plants14071079_

Round 1

Reviewer 1 Report (Previous Reviewer 2)

Comments and Suggestions for Authors

Thank you for your replies.

Unfortunately, I still miss the answers to 2 main aspects of my previous comments (metrological and analytical):  i.e., Comment 9, Comment 10,  and Comment 13.

Comment 9: I was referring to the saponification step itself. Do you think it is really needed or would ONLY extraction step be enough (without saponification). And I asked you as well whether you have tried to determine tocols in saponified and extracted AND in only extracted samples to justify the saponification step?

Comment 10: Thank you, but I still miss the estimation of the uncertainty due to this change (with all due respect to diminuishing harmful solvents (you could use n-heptane)).

Comment 13: Thank you for indicating the paper from the reference 37: Mišina et al., 2025 with detailed validation data, but it would be advisable to offer a generalized uncertainty estimation of the method within the present article as well.

Author Response

Dear Editor, dear Reviewers,

We sincerely thank you for all the comments, remarks, and suggestions that have contributed to enhancing the manuscript and its scientific quality. The manuscript and supplementary materials have been improved accordingly. Provided changes are marked in red font. For literature we used references manager software therefore changes are not highlighted.

Reviewer 1

Comment 0: Thank you for your replies. Unfortunately, I still miss the answers to 2 main aspects of my previous comments (metrological and analytical):  i.e., Comment 9, Comment 10,  and Comment 13.

Response 0: Thank you. We apologise our miss. Hoping this time we fully addressed to each comment.

Comment 9: I was referring to the saponification step itself. Do you think it is really needed or would ONLY extraction step be enough (without saponification). And I asked you as well whether you have tried to determine tocols in saponified and extracted AND in only extracted samples to justify the saponification step?

Response 9: Thank you; we now have a much clearer understanding of the question. No, we did not consider the extraction of tocochromanols without saponification. Our previous studies on seeds of grape (https://doi.org/10.1016/j.foodchem.2024.140913) and cranberry (https://doi.org/10.1016/j.scienta.2024.113107) demonstrated the highest extractability using the saponification protocol, while also indicating that the use of more environmentally friendly protocols (application of ethanol as solvent in presence of ultrasounds) is relatively comparable to saponification. We have discussed the advantages and disadvantages of using the saponification protocol and other techniques for the extraction of tocochromanols in detail in the review paper (https://doi.org/10.3390/molecules27196560). Since we did not conduct similar tests on Hypericum leaves, we have supplemented our findings with additional information to clarify and justify the use of the saponification protocol. Page 3, middle; page 13, bottom.

Comment 10: Thank you, but I still miss the estimation of the uncertainty due to this change (with all due respect to diminuishing harmful solvents (you could use n-heptane)).

Response 10: Thank you for the comment. Yes, it is true that n-heptane can be used instead of n-hexane. However, it is important to note the additional costs associated with this choice, as the price of n-heptane is higher than that of n-hexane. To minimize both costs and the quantity of harmful solvents used, we employ significantly reduced amounts of solvents—approximately seven times less than in the original protocol (50 mL vs. 7.5 mL). The method we utilized has been previously validated on apple seed samples (https://doi.org/10.1016/j.lwt.2014.05.006). To clarify this topic for readers, we have added in discussion a paragraph addressing saponification, extraction, and more environmentally and health-friendly protocols. A limitation of this paragraph is the extensive citation of our own works. However, the extraction processes, particularly the comparative analysis with the saponification protocol, and the discussion on the presence of potential tocopherol esters and non-extractable forms are relatively rare in the literature. We are opened and grateful for any suggestions. Page 3, middle.

Comment 13: Thank you for indicating the paper from the reference 37: Mišina et al., 2025 with detailed validation data, but it would be advisable to offer a generalized uncertainty estimation of the method within the present article as well.

Response 13: Thank you for the comment. The general uncertainty of the method was not calculated; however, we can add information about the method's repeatability and reproducibility. Page 14, middle.

Reviewer 2 Report (Previous Reviewer 3)

Comments and Suggestions for Authors

I acknowledge your response.

Author Response

Dear Editor, dear Reviewers,

We sincerely thank you for all the comments, remarks, and suggestions that have contributed to enhancing the manuscript and its scientific quality. The manuscript and supplementary materials have been improved accordingly. Provided changes are marked in red font. For literature we used references manager software therefore changes are not highlighted.

Reviewer 2

Comment 1: I acknowledge your response.

Response 1: Thank you.

Reviewer 3 Report (Previous Reviewer 4)

Comments and Suggestions for Authors

ABSTRACT

The abstract states "tocotrienols percentage was 13–32%" but does not specify whether this is by weight or volume, or the exact values for each species. This could provide more useful insight.

INTRODUCTION, RESULTS, DISCUSSION

While the article touches on the potential impact of drying methods on tocochromanol levels, it lacks a detailed mechanistic exploration of why certain drying methods might be more effective than others in preserving tocotrienols. A deeper understanding of the biochemical processes involved would strengthen the findings.

The absence of significant effects of drying methods on tocotrienol content is notable. However, the article does not sufficiently explain why this may be the case. Given the high susceptibility of tocotrienols to degradation, further exploration of factors like drying duration, temperature, or even the plant's physiological state would be useful.

MATERIAL AND METHODS

Some sections, such as the descriptions of drying methods (freeze-drying, microwave-vacuum drying, etc.), could be shortened or made more concise. The article repeats certain information multiple times (e.g., equipment names and conditions) which could be streamlined without losing clarity.

The saponification and extraction steps are described in a very detailed manner, but might feel overly repetitive. A more concise outline might suffice.

While various drying methods are described, the rationale behind choosing these specific techniques isn't fully explained. For example, why were these particular drying methods chosen over others? Including a brief justification based on prior research could help readers understand the choices better.

Similarly, the differences between drying in the presence of sunlight vs. darkness in the air-drying method could be discussed more, especially in terms of its impact on the overall goal of the study.

While equipment such as the FreeZone freeze-dry system and the microwave-vacuum system are mentioned, it would be beneficial to include more specifics about the machine capabilities (e.g., model numbers, key features) to ensure the reproducibility of the experiment.

The study could have benefited from addressing potential confounding variables that might influence tocopherol and tocotrienol content, such as environmental conditions, plant age, or seasonal variations.

Author Response

Dear Editor, dear Reviewers,

We sincerely thank you for all the comments, remarks, and suggestions that have contributed to enhancing the manuscript and its scientific quality. The manuscript and supplementary materials have been improved accordingly. Provided changes are marked in red font. For literature we used references manager software therefore changes are not highlighted.

Reviewer 3

Comment 1: ABSTRACT. The abstract states "tocotrienols percentage was 13–32%" but does not specify whether this is by weight or volume, or the exact values for each species. This could provide more useful insight.

Response 1: Thank you for the comment. Abstract has been improved in accordance with the guidelines. Page 1, middle.

Comment 2: INTRODUCTION, RESULTS, DISCUSSION. While the article touches on the potential impact of drying methods on tocochromanol levels, it lacks a detailed mechanistic exploration of why certain drying methods might be more effective than others in preserving tocotrienols. A deeper understanding of the biochemical processes involved would strengthen the findings. The absence of significant effects of drying methods on tocotrienol content is notable. However, the article does not sufficiently explain why this may be the case. Given the high susceptibility of tocotrienols to degradation, further exploration of factors like drying duration, temperature, or even the plant's physiological state would be useful.

Response 2: The explanation of tocotrienol stability was not a goal of this study; instead, it focused on their preservation, which was successfully achieved. Modifying drying parameters may not provide comprehensive insights into the underlying mechanisms of tocotrienol behavior, beyond their impact on preservation. This research effectively addressed the issue of α-T degradation in leaves and emphasized that the roles of other tocopherols and tocotrienols are still not well understood, necessitating further research. Consequently, the present study does not provide a detailed explanation of tocotrienol stability.

Comment 3: MATERIAL AND METHODS. Some sections, such as the descriptions of drying methods (freeze-drying, microwave-vacuum drying, etc.), could be shortened or made more concise. The article repeats certain information multiple times (e.g., equipment names and conditions) which could be streamlined without losing clarity. The saponification and extraction steps are described in a very detailed manner, but might feel overly repetitive. A more concise outline might suffice.

Response 3: Thank you, we appreciate suggestion. Modifications has been made in accordance with the guidelines. Page 12, bottom; page 13, top.

Comment 4: While various drying methods are described, the rationale behind choosing these specific techniques isn't fully explained. For example, why were these particular drying methods chosen over others? Including a brief justification based on prior research could help readers understand the choices better.

Response 4: Thank you, we appreciate suggestion. Modifications has been made in accordance with the guidelines. Page 12, middle (3.3.).

Comment 5: Similarly, the differences between drying in the presence of sunlight vs. darkness in the air-drying method could be discussed more, especially in terms of its impact on the overall goal of the study.

Response 5: Thank you, we appreciate suggestion. The influence of sunlight on the stability of tocopherols was not initially a primary objective of our study. However, after observing substantial losses of α-T, we were motivated to conduct further research under conditions with and without sunlight. A concise explanation of this aspect has been incorporated into the methodological section. Given the lack of similar studies in the literature, we have endeavored to provide a thorough discussion on this topic. Page 13, middle.

Comment 6: While equipment such as the FreeZone freeze-dry system and the microwave-vacuum system are mentioned, it would be beneficial to include more specifics about the machine capabilities (e.g., model numbers, key features) to ensure the reproducibility of the experiment.

Response 6: Thank you for the comment. We provided all the details in the previous version. We improved by adding „4.5 L” for the FreeZone freeze-dry system. We do not think it is a proper way to include such information in the main text. In supplementary materials can be seen the used equipment. We added also photos of the required details.

Comment 7: The study could have benefited from addressing potential confounding variables that might influence tocopherol and tocotrienol content, such as environmental conditions, plant age, or seasonal variations.

Response 7: Thank you for your suggestion. Modifications has been made in accordance with the guidelines. Page 6, middle.

Reviewer 4 Report (New Reviewer)

Comments and Suggestions for Authors
  • Improve abstract
  • Introduction: Strengthen the connection with existing research on the optimization of drying methods, even for other plants.
  • There is Tocochromanol Profile only for the Freeze-Dried method. And the other drying methods?
  • Line 137-141: Include tocotrienol concentrations from other studies for more clarity
  • correct Apiaceae and other families in italic form throughout manuscript (Apiaceae)
  • Clarify the contribution of your study to industrial or academic applications.

Suggestion : comparison of the results obtained with the contents of the fresh plant.

Author Response

Dear Editor, dear Reviewers,

We sincerely thank you for all the comments, remarks, and suggestions that have contributed to enhancing the manuscript and its scientific quality. The manuscript and supplementary materials have been improved accordingly. Provided changes are marked in red font. For literature we used references manager software therefore changes are not highlighted.

Reviewer 4

Comment 1: Improve abstract.

Response 1: Thank you, we appreciate suggestion. Modifications has been made in accordance with the guidelines.

Page 1, middle.

Comment 2: Introduction: Strengthen the connection with existing research on the optimization of drying methods, even for other plants.

Response 2: Thank you for the comment. Here lies the challenge, as generally, is lack of studies on drying the leaves and method impact on tocochromanols content. This scarcity makes it challenging to engage in discussions regarding leaf drying. Despite these challenges, some adjustments were made in the introduction. This situation highlights the need for more research on the drying processes for leaves, particularly focusing on how different methods might impact the stability of tocopherols and tocotrienols. Page 2, bottom; page 3, top.

Comment 3: There is Tocochromanol Profile only for the Freeze-Dried method. And the other drying methods?

Response 3: Thank you for your comment. The tocochromanol profile remained unaffected by the type of drying method employed. Consequently, the discussion centered on freeze-drying method, which is known to have the higest preservation of tocochromanols during the drying process. In the following paragraph, a comparative analysis was conducted to assess the tocochromanols content in leaves subjected to various drying techniques. This comparison aimed to elucidate how different drying methods impact the retention of these bioactive compounds.

Comment 4: Line 137-141: Include tocotrienol concentrations from other studies for more clarity

Response 4: Thank you, we appreciate suggestion. Modifications has been made in accordance with the guidelines. Page 5, bottom.

Comment 5: correct Apiaceae and other families in italic form throughout manuscript (Apiaceae)

Response 5: Thank you for the comment, however, according to the Angiosperm Phylogeny Group classification for the orders and families of flowering plants, the names of families are not written in italic style. Please see:

https://doi.org/10.1046/j.1095-8339.2003.t01-1-00158.x

https://doi.org/10.1111/j.1095-8339.2009.00996.x

https://doi.org/10.1111/boj.12385

Comment 6: Clarify the contribution of your study to industrial or academic applications.

Response 6: Thank you for the comment. We believe that this issue we already well highlighted in the conclusion enough.

Comment 7: Suggestion: comparison of the results obtained with the contents of the fresh plant.

Response 7: Thank you for the comment. We did not conduct investigations on fresh leaves because this type of study is susceptible to several errors. The rapid dehydration of leaves, enzymatic activity, and difficulties in preparing these samples for analysis pose significant challenges.

Round 2

Reviewer 1 Report (Previous Reviewer 2)

Comments and Suggestions for Authors

Dear authors!

Thank you for your additional responses.

It seems you have covered all of my doubts. I still think the uncertainty budget should be included in every scientific work that deals with (chemical) measuring, but let's be satisfied with the repeatability and reproducibility for now.

This manuscript is a resubmission of an earlier submission. The following is a list of the peer review reports and author responses from that submission.

Round 1

Reviewer 1 Report

Comments and Suggestions for Authors

The scope of the experiment is very narrow. Only one determination was made on a very small set of samples. I do not see any novelty in this work. Besides, statistical analysis with less than 20 samples should be based on non-parametric tests. I definitely do not recommend for publication.

Reviewer 2 Report

Comments and Suggestions for Authors

Leva Valäte et al. - Referee's comments

Please try to improve the syntax in the line(s) 84-85, 90-94, 98-101, 113-117, 143-144, 157-163, 166-167,  300-301, 309-310, 312-315, 323-324, 325-326, 342-344, 347-348, 350-351, 357-361

Comment 1: line 13: typo in “in-frared-oven-“

Comment 2: lines 107-109: inaccurate expression – only 2 T3 homologues (delta and gamma are present) – please define which T3 are present

Comment 3: lines 111-112 (same as in comment 2)

Comment 4: lines 144-147: inaccuracy – you have not detected beta-T3. Please correct.

Comment 5: line 175: typo in “air-dring”

Comment 6: lines 183-185: please define which species.

Comment 7: lines 193-194 & Table 1: please give some additional information & explain in more depth the presented data

Comment 8:  lines 271-274. How the T & T3 standards were used or prepared to fulfil the determination range is unclear. It stands primarily for the T3 standards – the Cayman Chemical web page lists these T3 standards as solutions in ethanol! As can be understood from their Production information page (e.g. delta-T3 https://cdn.caymanchem.com/cdn/seawolf/insert/10008513.pdf ), they are QAUALI and NOT QUANTI standards. Please comment.

Comment 9: why saponification and not only extraction? Have you tried to determine tocols in saponified and only extracted samples to justify the saponification step?

Comment 10: line 347: why only 0.1 g? How does this mass influence the final uncertainty of the determination?

Comment 11: lines 354-357: please use a more appropriate verb than “incubate”

Comment 12: have you determined the recovery of the extraction?

Comment 13: I miss some analytical-quantitative and validation data in paragraph 3.5. (injection volume, repeatability, reproducibility, linearity, calibration standards range, accuracy, LOD, LOQ, uncertainty, …). I guess this was already covered in GórnaÅ›, Siger, Czubinski, et al. (2014), but it would be proper to mention it in this paper and point to eventual differences.

Comment 14: Maybe using isomer instead of homologue would be more appropriate—just a suggestion (I see in reference [31] that you used the term “homologue” as well).

Comment 15: How was the identification of tocols performed?

Comment 16: Don’t you find it strange that the prevalent tocopherol is alfa and the prevalent tocotrienol is delta? Please comment on this.

Reviewer 3 Report

Comments and Suggestions for Authors

The plant material used in this paper does not contain α-T3. In this paper, a significant yield reduction due to air-dry was observed for α-T only. b-T, d-T and g-T were not. Extraction modeling experiments with α-T3 using standard compounds should be carried out.

Reviewer 4 Report

Comments and Suggestions for Authors

ABSTRACT

Some sentences are awkwardly phrased or overly complex (e.g., "in the genus Hypericum the situation is at least not clear").

Certain technical details could be presented more fluidly.

The introduction could be condensed, and the focus on findings and implications could be expanded.

The term "α-T" should be spelled out at its first mention for clarity.

Phrases like "the situation is at least not clear" and "domination of δ-T3 and γ-T3" need rephrasing for precision.

INTRODUCTION

The introduction is dense and could be divided into clearer sections. For example:

      • A separate section for botanical background.
      • A section for pharmacological importance and current knowledge gaps.
      • A focused section on the rationale for studying drying methods and their impact on tocotrienols.

Repetition of certain ideas, such as the medicinal importance of H. perforatum and the lack of research on tocotrienols, could be reduced.

The discussion of tocotrienols in other plant species, while informative, takes up significant space without clear ties to Hypericum until later in the paragraph. Focus on Hypericum earlier.

The mention of "natural drying methods" and other techniques could be streamlined to avoid tangential details.

Phrases such as "is certainly already higher taking at least the recent discovery..." could be rephrased for smoother readability.

Avoid redundant phrases like "during the last three decades" and "recently."

Introduce the study aim earlier in the introduction and reiterate it with a stronger transition to the experimental section .

RESULTS AND DISCUSSION

There is some redundancy in presenting species comparisons and the impact of drying methods (e.g., overlap between the main text and tables). Minimize repetition by summarizing key results in text and referencing figures/tables for detailed data.

The section identifies differences but does not delve into the biochemical or physiological reasons for variations in tocopherol/tocotrienol profiles. Discuss possible biosynthetic pathways or environmental factors influencing tocopherol/tocotrienol variation.

The narrative around chromatogram interpretations (e.g., Figure 1) could be clearer to avoid misconceptions, such as overestimating tocotrienol dominance due to chromatogram peak heights. Address the potential misinterpretation of chromatographic results directly in the text and ensure all figures are annotated appropriately.

Highlight practical implications of findings, such as recommended drying methods for preserving specific tocopherol/tocotrienol homologues.

Include a brief comparison with other genera or previously studied Hypericum species to place findings in a broader biological and industrial context.

MATERIALS AND METHODS

Excessive detail on certain aspects, such as the soil analysis report, could overwhelm readers. Include only essential soil parameters in the main text and move the full soil analysis details to supplementary materials.

Redundant information, such as repeating the drying conditions for the October experiment, can be streamlined. Consolidate the description of similar procedures.

There is inconsistency in how biological replicates are referred to (e.g., "three biological replications" vs. "biological replicates"). Use standardized terminology throughout the section.

Some descriptions (e.g., microwave-vacuum drying optimization) are unclear about how the conditions were determined. Briefly state how optimization was achieved or reference a prior publication/method.

The description of the statistical methods does not specify how assumptions (e.g., normality, homogeneity) were tested. Add a sentence explaining how these assumptions were verified.

Some methods, such as the air-drying technique, describe conditions without justifying their selection. Briefly explain the rationale behind the chosen drying methods (e.g., relevance to real-world practices or prior studies).

The mention of Tukey's post-hoc test does not specify for which factors the test was applied (e.g., species, drying method). Clarify how the two-factor ANOVA and post-hoc tests were used.

Several references to figures and tables in the supplementary materials interrupt the flow. Ensure these references are concise, and all supplementary material is appropriately organized.

CONCLUSIONS

The phrasing is somewhat dense and could be reorganized for better readability.

Provide more context for chemotaxonomic tools. Briefly describe their role in phytochemical analysis and potential applications.

Acknowledge any constraints in methodology or scope and how they might influence the interpretation of results.

Propose expanded studies on additional species or regions and optimization of drying techniques for industrial scalability.

REFERENCES

There are five self-citations that do not include previous original results and are not part of the discussion, but rather appear in the introduction. These should be revised.

Comments on the Quality of English Language

The English is mostly fine, but some sentences are overly long or complicated, and there are a few instances of awkward grammar.